# Accuracy of the diagnosis of pneumonia in Canadian pediatric emergency departments: A prospective cohort study

**Joan L. Robinson**[1]*, **James D. Kellner**[2], **Jennifer Crotts**[2], **Gabino Travassos**[2], **Guanmin Chen**[3], **Valerie G. Kirk**[2], **Martin Pusic**[4], **Martin Reed**[5], **Sarah Reid**[6], **Michael Weinstein**[7], **Ravi Bhargava**[1], **Maala Bhatt**[6], **Kathy Boutis**[7], **Sarah Curtis**[1], **Serge Gouin**[8], **Tim Lynch**[9], **Richard van Wylick**[10], **David W. Johnson**[2]

**1** Faculty of Medicine & Dentistry, University of Alberta, Edmonton, Alberta, Canada, **2** Department of Pediatrics, Cumming School of Medicine, University of Calgary, Calgary, Alberta, Canada, **3** Alberta Health Services, Alberta, Canada, **4** Pediatrics, Harvard Medical School, Boston, MA, United States of America, **5** University of Manitoba, Winnipeg, Manitoba, Canada, **6** Department of Pediatrics, University of Ottawa, Ottawa, Ontario, Canada, **7** University of Toronto, Toronto, Ontario, Canada, **8** Centre Hospitalier Universitaire Sainte-Justine, Montreal, Quebec, Canada, **9** Department of Pediatrics, Schulich School of Medicine, Western University, London, Ontario, Canada, **10** Queen's University, Kingston, Ontario, Canada

* jr3@ualberta.ca

**Data Availability Statement:** The data underlying the results presented in the study are available from Biostatistical Support, Alberta Children's

## Abstract

### Background

The diagnosis of pediatric pneumonia and determination of the likely pathogen are complicated as the clinical picture overlaps with other respiratory illnesses, interpretation of radiographs is subjective, and laboratory results are rarely diagnostic. This study was designed to describe the relative rates of bacterial and viral pneumonia in the pediatric Emergency Department (ED), determine the accuracy of pediatric ED physicians' ability to diagnose pneumonia and distinguish bacterial from viral etiology, and to determine clinical and laboratory predictors of bacterial pneumonia.

### Methods

Children 3 months to 16 years of age presenting to seven Canadian pediatric EDs before the COVID-19 pandemic with fever and cough who had a chest radiograph performed for possible pneumonia were enrolled and underwent standardized clinical investigations. An expert panel was convened and reached a Consensus Diagnosis of typical or atypical bacterial pneumonia, viral pneumonia or not pneumonia for each case.

### Results

The expert panel assessed 247 cases with the Consensus Diagnosis being typical bacterial pneumonia (N = 44(18%)), atypical bacterial pneumonia (N = 18(7%)), viral pneumonia (N = 46(19%)) and no pneumonia (N = 139(56%)). Treating ED physician diagnoses were typical bacterial pneumonia (N = 126(51%)), atypical bacterial pneumonia (N = 3(1%)), viral pneumonia (N = 10(4%)) and no pneumonia (N = 108(44%)) with low agreement between a diagnosis of bacterial pneumonia by the ED physician and the panel's Consensus Diagnosis

Hospital Research Institute, University of Calgary, Calgary, Canada. [r4k@ucalgary.ca].

**Funding:** Funding for Pneumonia Prospective Cohort Study • Canadian Institutes for Health Research Team Grant, Funded from April 2006-March 2011, Grant title: 'Improving outcomes for ill and injured children in emergency departments'. Principal Investigator – Terry Klassen; Co-Investigators (and pneumonia project leads) – David Johnson and Tim Lynch. Can$4.8 million funded seven large multi-centre projects, one of which focused on three distinct pneumonia studies (systematic review, practice variation study and prospective cohort study) • Alberta Children's Hospital Foundation project grant, Funded from April 2009-April 2010, Grant title: 'Accuracy of metabolomics for diagnosing pediatric pneumonia'. Principal Investigator – Jim Kellner; Co-Investigator – David Johnson. Can$50,000 • Alberta Lung Association project grant. Funded from April 2009-April 2010, Grant title: 'Accuracy of metabolomics for diagnosing pediatric pneumonia'. Co-investigator. Principal Investigator – Jim Kellner; Co-Investigator – David Johnson. Can $30,000 The funders had no role in study design, data collection and analysis, decision to publish, or preparation of the manuscript.

**Competing interests:** The authors have declared that no competing interests exist.

(Kappa 0.15 (95% CI 0.08, 0.21)). Cut off values that predicted bacterial pneumonia as the Consensus Diagnosis were ESR $\geq$ 47 mm/hour, CRP $\geq$ 42 mg/L and procalcitonin $\geq$0.85 ng/m. Age greater than 5 years and cough for 5 or more days also predict bacterial pneumonia.

## Conclusion

In this cohort, pediatric ED physicians over-diagnosed typical bacterial pneumonia and underdiagnosed viral and atypical bacterial pneumonia. Bacterial pneumonia is most likely in children over 5 years of age, with cough for 5 or more days and/or with elevated inflammatory markers.

## Introduction

In high-income countries, pneumonia results in significant childhood morbidity and, rarely, mortality. Hospitalization rates in the United States are estimated to be 15.7–22.5 per 100,000 children [1]. Establishing the suspected pathogen leads to appropriate treatment but is challenging. Typical and atypical bacteria or viruses cause the vast majority of cases, with viruses predominating in young children [2]. For typical bacterial pneumonia, antibiotics appear to hasten resolution, prevent almost all deaths, and may decrease the incidence of complications including empyema or abscess. Atypical pneumonia is commonly self-resolving, but effective antibiotics hasten resolution [3]. Antibiotics play no role in the management of viral pneumonia.

Given the absence of a proven reference standard [4], clinicians struggle with the accurate diagnosis of bacterial pneumonia. The presenting symptoms and physical examination findings overlap with those of upper respiratory viral infections, bronchiolitis, asthma and viral pneumonia [5]. Beyond history and physical examination, a common investigation includes chest radiograph (CXR), yet even pediatric radiologists may disagree about the presence of pneumonia or consolidation [6]. Less often, laboratory and microbiologic tests are performed but these can be distressing for the child and are time consuming. As a result, these tests are often not used. C-reactive protein (CRP) and procalcitonin (PCT) are more useful biomarkers than white blood cell counts in differentiating bacterial pneumonia from other causes of respiratory distress, but sensitivity and specificity are only approximately 70% and 65% for CRP or PCT respectively [7]. In Canadian pediatric EDs, PCT is often not readily available. Viruses and bacteria detected from upper respiratory tract specimens may or may not be the cause of pneumonia. Lower tract specimens can be contaminated from the upper tract and can only be obtained if a child requires intubation.

There is evidence that expert panels using formalized consensus methods provide accurate diagnoses comparable to that of reference standards, are highly reproducible and are increasingly used in lieu of definitive reference standards [8–11]. For these reasons we utilized an expert panel and a formal method for reaching consensus to achieve the following study objectives: 1) describe the occurrence of viral and bacterial pneumonia in children presenting to the Emergency Department (ED) with suspected pneumonia, 2) determine the accuracy of ED physicians in diagnosing pneumonia and distinguishing bacterial from viral pneumonia, and 3) determine the accuracy and reliability of individual clinical findings and common laboratory tests for predicting bacterial pneumonia.

## Materials and methods

### Study design and setting

We used a prospective cohort design, with standardized collection of history, physical examination findings, laboratory and microbial testing results, CXR reports and telephone follow-up assessments after ED discharge for children clinically suspected to have pneumonia. A panel of experienced pediatric sub-specialists reviewed each case using their individual clinical expertise to achieve consensus. The recommendations of a systematic review of studies using expert panels to define a reference standard were incorporated [9]. The *Strengthening the Reporting of Observational Studies in Epidemiology* (STROBE) Statement guidelines for reporting observational studies were followed [12].

### Inclusion criteria

Eligible patients were those aged 3 months to 16 years presenting to one of seven Canadian pediatric EDs from August 1, 2008 through April 30, 2011 with both fever (either on history or on physical exam) and cough for whom the ED physician ordered a CXR.

### Exclusion criteria

Patients were excluded if: the presumptive diagnosis was croup, bronchiolitis, or moderate or severe asthma (defined as treatment with more than two bronchodilator aerosols in the ED); if the child had a significant chronic comorbidity; or a history of pneumonia confirmed by CXR in the preceding 8 weeks, or receiving antibiotic treatment between 6 hours and 14 days prior to arriving in the ED. Other exclusions included significant language barrier, previous enrolment in the study, or anticipated inability to complete telephone follow-up.

### Ethics approval and study enrollment

The study was approved by the research ethics board of all seven sites. Trained study nurses were on-call at each site, typically between 20 and 40 hours per week and were notified of potentially eligible patients by ED personnel. Study nurses determined whether patients met all eligibility criteria, obtained written informed consent from parents and ensured standardized data collection.

### Data collection

Demographic variables and clinical signs and symptoms at presentation and during the course of ED assessment and the final diagnosis by the ED physician were documented. Complete blood count (CBC), erythrocyte sedimentation rate (ESR), CRP and blood culture were analyzed locally at each site. Serum for PCT and nasopharyngeal swabs (NPS) were transported on dry ice by express delivery to the central study site and stored at minus 80 centigrade until processing. NPS were cultured for pneumococcus, *Haemophilus* species, *Staphylococcus aureus*, group A streptococcus, *Moraxella catarrhalis* and *Bordetellae* and molecular detection using a validated technique was performed for influenza, adenovirus, bocavirus, endemic coronaviruses, enterovirus, human metapneumovirus, parainfluenza virus, respiratory syncytial virus (RSV), rhinovirus), *Chlamydiae*, and *Mycoplasma* [13].

Parents were asked to maintain a fever log for seven days following the ED visit. Specifically, they were asked to periodically assess their child for fever by palpation, and if the child was noted to be warm, measure and log their axillary temperature using the provided digital thermometer (*AccuflexPro* digital, *Physiologic®*). Parents were contacted for follow-up assessments by telephone at day 7 and day 28 to ask about return visits to a physician or to an ED,

complications including hospital admission, empyema or abscess formation, therapeutic changes, and results of the fever log (day 7 only).

## Interpretation of CXRs

CXRs were interpreted by the ED physician and radiologist at the site where the patient was enrolled. At the completion of the study, all CXRs were also interpreted by the study radiologist (a senior pediatric radiologist) using the *Standardization of Interpretation of Chest Radiographs for the Diagnosis of Pneumonia in Children* (WHO/V&B/01.35/2001).

## Consensus (expert) panel

A consensus panel was assembled, consisting of the following seven members all of whom are board certified in their specialty and sub-specialty: two pediatric ED physicians, two pediatric infectious disease physicians, one pediatric radiologist, one pediatric respirologist and one pediatric hospitalist. Panelists had a median 26 years of clinical experience, ranging between 15 and 47 years. Each panelist used their clinical expertise to interpret the significance of laboratory test results and chest radiographs; no interpretation standards were imposed on panelists.

## Data visualization by panelists

A customized secured-access web portal was developed which allowed panelists to view all data asynchronously including high quality digital CXR images (Figs 1–6); the web portal was hosted by the central study site. Panelists made their diagnoses three separate times per case, based on three successive waves of data: 1) immediate clinical information: history, vital signs and other findings on physical examination (PE); 2) clinical plus immediate laboratory information: CXR image and CBC and differential results; and 3) all other data including follow-up PCT, ESR, CRP, blood culture and NPS results, interpretation of the CXR by a) site ED physician, b) site radiologist, and c) study radiologist, discharge information including patient disposition and treatment at discharge, results of follow-up assessment at day 7 including graph of the fever pattern and results of the follow-up telephone assessment at day 28 (Figs 1–6).

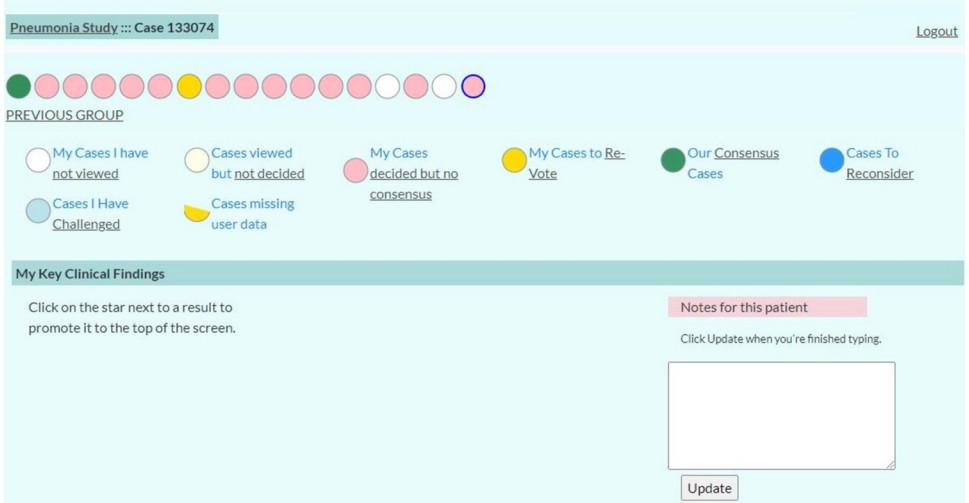

**Fig 1. Home screen.** Screenshot of Home Screen which summarized case status and allowed panelists to navigate between cases.

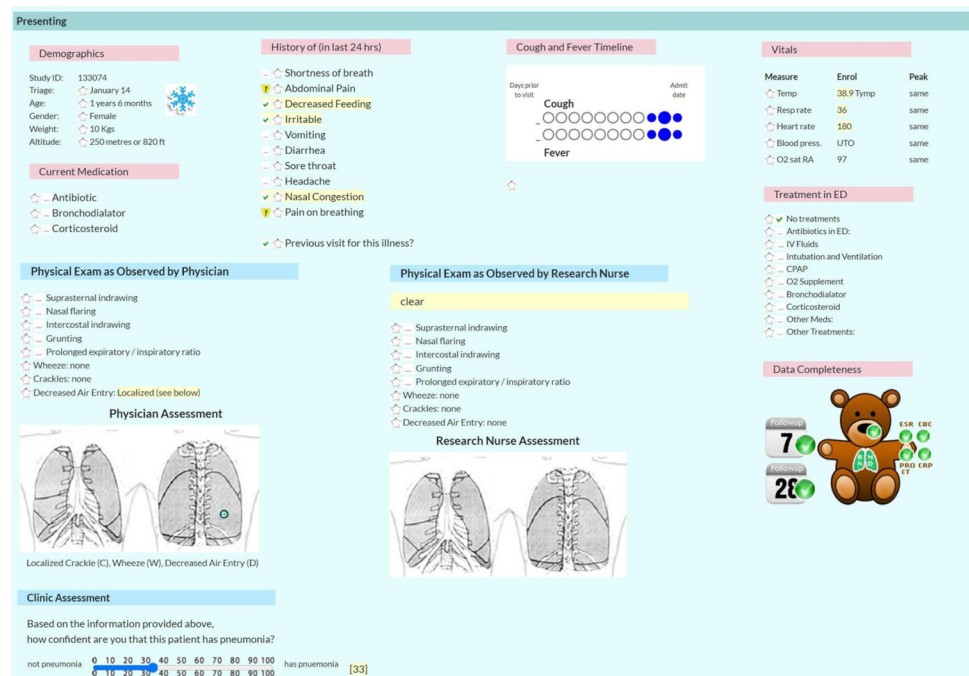

**Fig 2. Clinical assessment.** Screen shot of history, physical examination and clinical assessment.

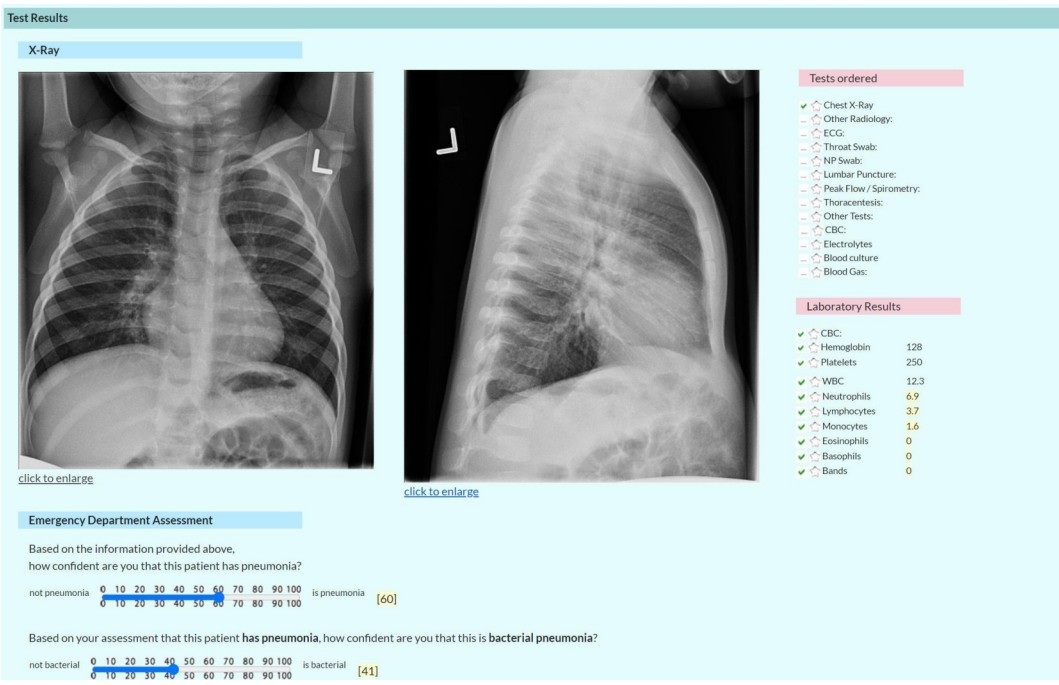

**Fig 3. Laboratories, CXR and ED assessment.** Screenshot of Test Results, CXR image, and Emergency Department Assessment.

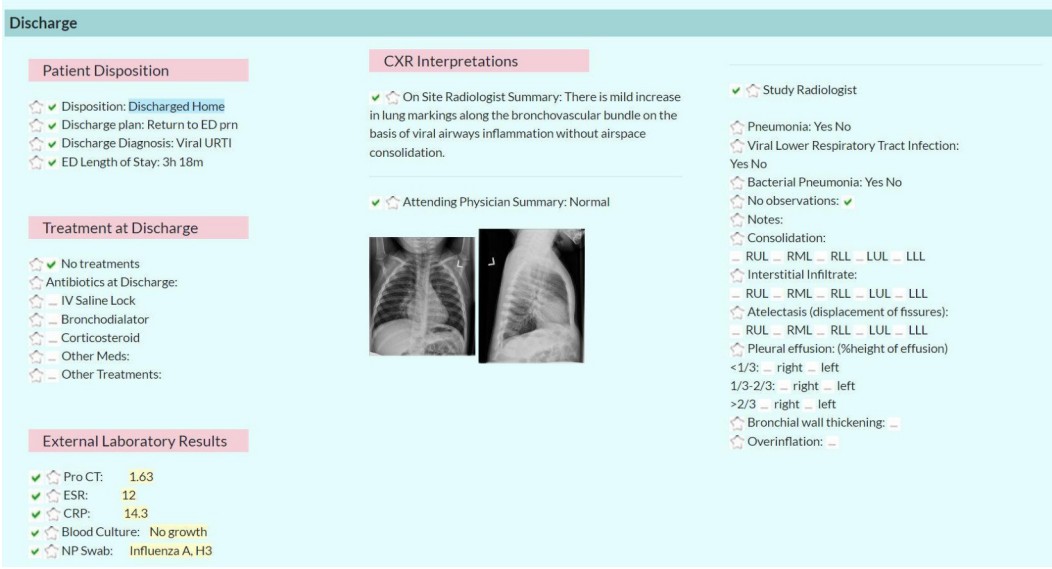

**Fig 4. Discharge information, follow-up laboratories and CXR interpretations.** Screenshot of Discharge information, follow-up laboratory results, and various CXR interpretations.

## Process for panelists to select their diagnosis and indicate their degree of certainty

After reviewing each of the three waves of information, panelists adjusted an onscreen slider ranging between 0 and 100 with 0–49 indicating they chose 'no pneumonia' and 51 to 100 indicating they chose 'pneumonia'. Selecting a number approaching '0' or '100' indicated they were very certain of their diagnosis whereas choosing a number approaching 50 indicated they

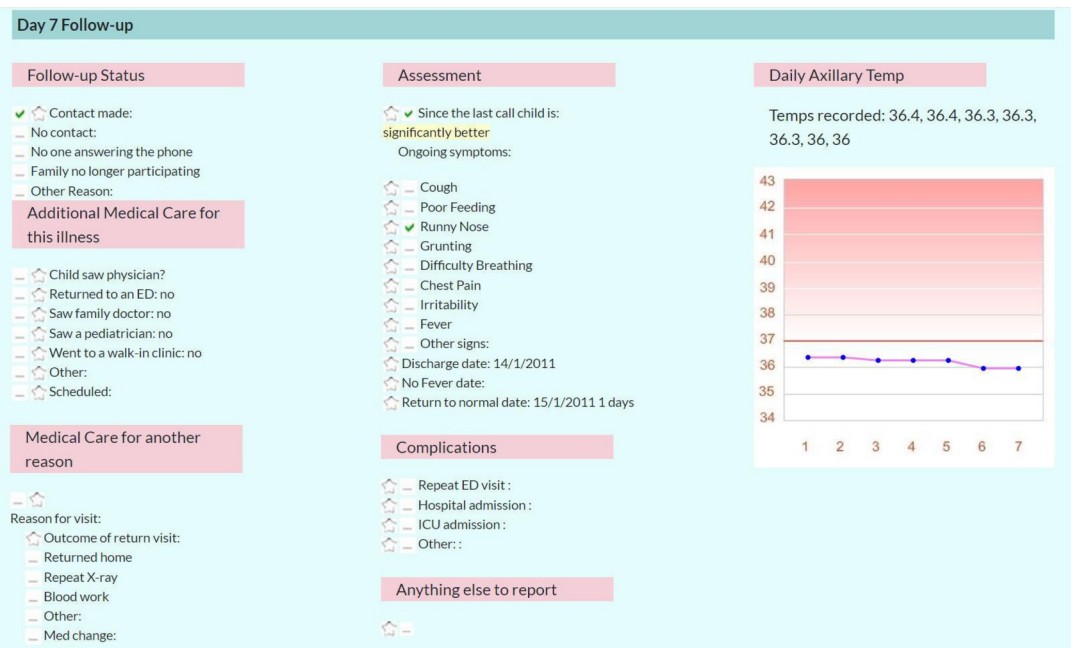

**Fig 5. Day 7 follow-up.** Screenshot of Day 7 Follow-up.

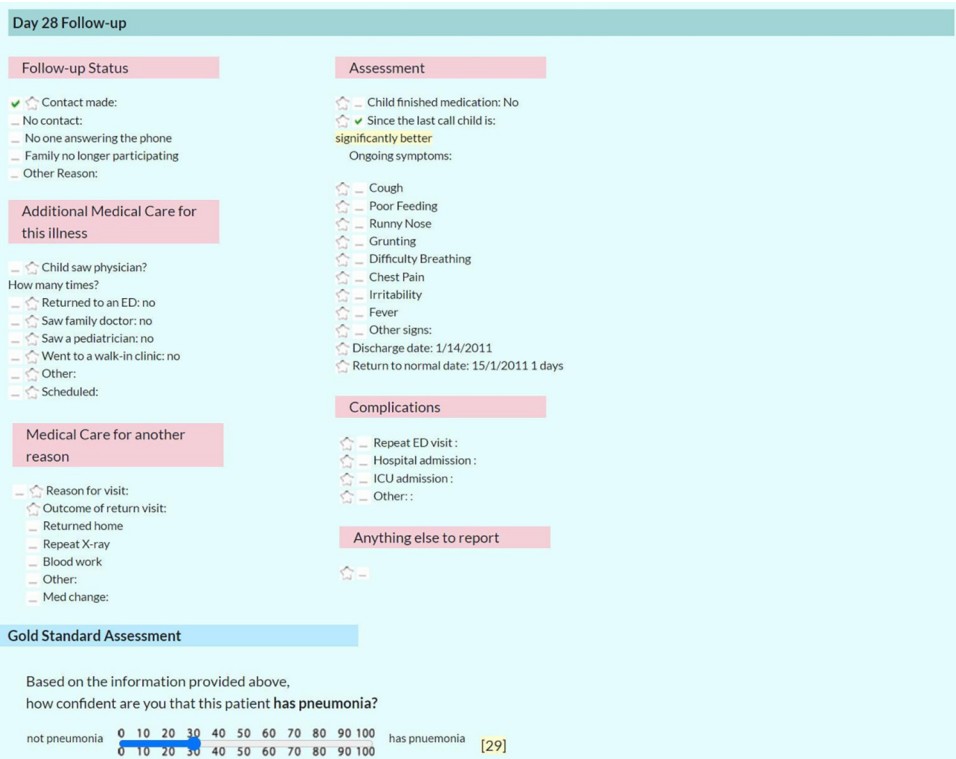

**Fig 6. Day 28 follow-up and full data assessment.** Screenshot of Day 28 Follow-up and Full-Data Assessment [labeled in screenshot "Gold Standard Assessment"] which includes panelist certainty (expressed in 1 to 100%).

were very uncertain. If they selected 'pneumonia', they were presented with a second slider that was identical except the polar choices were 'viral pneumonia' and 'bacterial pneumonia'. After reviewing all data from the three phases, panelists who selected bacterial pneumonia were presented with an additional slider identical except the polar choices were 'atypical bacterial pneumonia' and 'typical bacterial pneumonia'.

The panelist diagnosis selected following review of the first data set (history and PE findings) is referred to as the *Immediate Clinical Assessment*. The panelist diagnosis selected following review of the second data set (CXR image and CBC) is referred to as the *Full ED Assessment*. The panelist diagnosis selected following review of third and final data set (inflammatory markers and microbiological results, and discharge and follow-up information) is referred to as the *Full Follow-up Assessment*.

### Process for achieving panel's consensus diagnosis

After all panelists chose their *Full Follow-up Assessment* (their diagnosis based on all three data sets) for a case, if the majority of the seven panelists (i.e. minimum of four members) chose the same diagnosis (no pneumonia, viral pneumonia, atypical bacterial pneumonia, or typical bacterial pneumonia), this was considered the *Consensus Diagnosis*. If not, consensus was "forced" as follows. If the majority selected pneumonia but disagreed on the type of pneumonia, those who selected "no pneumonia" were required to choose between the three types of pneumonia and change the slider that recorded their degree of certainty. If after this re-vote, four or more members chose the same type of pneumonia, this was considered the *Consensus Diagnosis*. If not, then for example if the majority chose typical or atypical bacterial pneumonia, those who chose viral pneumonia were required to choose between typical and atypical bacterial

pneumonia. At any point in this process, panelists who strongly disagreed with the choice they were being "forced" to make could send a written argument to all panelists, asking those in the majority to reconsider their choice. If after seven days the panelists in the majority did not change their diagnosis, the dissenting minority panelist was "forced" to conform with the majority diagnosis. The diagnosis (no pneumonia, viral pneumonia, atypical bacterial pneumonia, or typical bacterial pneumonia) eventually selected by four or more panelists was the *Consensus Diagnosis*, and any panelists not in agreement were "forced" to select this diagnosis. However, if they still disagreed, they were able to reflect this by indicating a low degree of certainty by moving their slider toward 50%. Certainty was determined by calculating the percentage difference between the chosen value and 50.0%, then this number was multiplied by 2. This allowed certainty to range from a minimum of 0 and a maximum of 100. For example, if a panelist chose 51.0%, then the calculated certainty would be 2.0%.

### Estimation of panel's aggregate immediate clinical, full ED and full follow-up assessments

In order to assess agreement between the Panel's *Consensus Diagnosis* and the Panel's aggregate *Immediate Clinical, Full ED and Full Follow-up Assessments*, the latter was represented by determining the majority vote of the seven panelists combined with their diagnostic certainty.

### Sample size

Based on a pediatric ED study enrolling children with suspected pneumonia with fever and cough which reported 37% had focal infiltrates on CXR [14], we conservatively estimated that 30% of enrolled children enrolled in our study would have a Consensus Diagnosis of bacterial pneumonia. As our funding for this study limited the number of patients we could enroll to 250, we anticipated approximately 75 children (95% CI 60, 90) would have a consensus diagnosis of bacterial pneumonia.

### Data analysis

Descriptive statistics (i.e. frequencies, percentage, means, and standard deviations, IQR) were used to describe demographic, clinical characteristics, examination findings and laboratory results for overall study population and stratified by the *Consensus Diagnosis*. Pneumonia was classified as any pneumonia (either viral or bacteria), any bacterial pneumonia (either typical or atypical bacteria), or typical bacteria pneumonia.

Patient findings were compared between the consensus pneumonia types using chi-square test for proportions, Students t-test for means, or Kruskal-Wallis test for medians. Diagnostic agreement (kappa, sensitivity, specificity, positive predictive value (PPV), negative value (NPV)) between *Consensus Diagnosis* and the *Site ED Physician Diagnosis* were calculated for overall study population.

For ESR, CRP and PCT, optimal cut-points were determined using area under the Receiver Operation Curve (ROC) and logistic regression modeling. These cut-points were then used to predict each laboratory test's diagnostic accuracy for bacterial pneumonia.

We used logistic regression modeling to explore the association between clinal characteristics and laboratory results, and pneumonia types. To explore the accuracy of clinical and laboratory variables for predicting a diagnosis of pneumonia, area under the ROC were estimated for each variable in the logistic regression model. An alpha of 0.05 was used to determine statistical significance. A *Sanky flowchart* was used to show the diagnostic comparison between the site ED physician and Consensus Diagnoses. All analyses were performed using SAS statistical software (*SAS version 9.4, SAS Institute*, Cary, North Carolina).

**Table 1. Characteristics of children assessed in the ED with cough and fever who had a CXR performed.**

| | | | No pneumonia[1] (N = 139) | Bacterial pneumonia[1] (N = 62) | Viral pneumonia[1] (N = 46) | Total (N = 247) |
|---|---|---|---|---|---|---|
| Age category (years) | <1 | n (%) | 12 (8.6) | 4 (6.5) | 9 (19.6) | 25 (10.1) |
| | 1–5 | n (%) | 91 (65.5) | 28 (45.2) | 37 (80.4) | 156 (63.2) |
| | 6–9 | n (%) | 19 (13.7) | 16 (25.8) | 0 (0.0) | 35 (14.2) |
| | 10–16 | n (%) | 17 (12.2) | 14 (22.6) | 0 (0.0) | 31 (12.6) |
| Age (years) | | Mean (SD) | 4.3 (3.9) | 6.2 (4.1) | 2.5 (1.5) | 4.5 (3.9) |
| O2 sat room air in ED (%) | | Mean (SD) | 96.5 (2.6) | 96.1 (3.2) | 95.4 (3.2) | 96.2 (2.9) |
| Respiratory rate (breaths per minute) | | Mean (SD) | 31.6 (10.6) | 31.4 (12.3) | 37.0 (9.8) | 32.5 (11.1) |
| Heart rate (beats per minute) | | Mean (SD) | 134.4 (30.8) | 128.8 (28.4) | 145.0 (20.6) | 135.0 (28.9) |
| Temperature in ED (°C) | | Mean (SD) | 37.8 (1.0) | 37.9 (1.0) | 38.2 (1.1) | 37.9 (1.1) |
| Fever (≥38°C) in ED | | n (%) | 58 (41.7) | 32 (51.6) | 23 (50.0) | 113 (45.7) |
| Fever duration (days) | | Median (IQR) | 3 (1, 5) | 3 (2, 5) | 3 (1, 5) | 3 (1, 5) |
| Fever (≥ 5 days) | | n (%) | 36 (25.9) | 19 (30.6) | 12 (26.1) | 67 (27.1) |
| Cough duration (days) | | Median (IQR) | 4 (2, 7) | 6 (3, 7) | 5 (3, 7) | 5 (3, 7) |
| Cough (≥5days) | | n (%) | 61 (43.9) | 39 (62.9) | 27 (58.7) | 127 (51.4) |
| Season | Apr-Jun | n (%) | 18 (12.9) | 8 (12.9) | 8 (17.4) | 34 (13.8) |
| | Jan-Mar | n (%) | 61 (43.9) | 24 (38.7) | 15 (32.6) | 100 (40.5) |
| | Jul-Sep | n (%) | 8 (5.8) | 7 (11.3) | 1 (2.2) | 16 (6.5) |
| | Oct-Dec | n (%) | 52 (37.4) | 23 (37.1) | 22 (47.8) | 97 (39.3) |

CXR–chest radiograph; ED–emergency department; IQR–inter-quartile range; SD—standard deviation

[1] This diagnosis was based on the Consensus Diagnosis of an expert panel.

## Results

### Characteristics of enrolled children

The health records of 2,634 children were screened of whom 1294 met study eligibility. Of these, a study nurse was not available or contacted for 652. Of the 642 patients whose parents were approached about the study, 373 refused consent, resulting in the enrollment of 269 children. Of these children, 10 withdrew, 8 had major protocol violations, and another 4 had insufficient data to be included in the consensus panel review, leaving 247 cases. Of these 247 cases, 11 could not be contacted for follow-up and another 31 did not complete the fever diary but provided the other follow-up data. The number enrolled at each of the seven sites ranged from 4 to 81 (median 30). *Consensus Diagnosis* was bacterial pneumonia in 62 cases (25%), viral pneumonia in 46 cases (19%) and no pneumonia in 139 cases (56%) (Table 1). The majority were 1 to 5 years of age (63%). Fever had been present for median 3 days and cough for median 5 days on presentation, with 80% presenting October through March.

### Laboratory results

See Table 2 for a summary of laboratory and microbiological testing results. At least one virus was detected from 153 of 247 cases (62%), including 17 of 62 cases (27%) with a *Consensus Diagnosis* of bacterial pneumonia.

**Table 2. Laboratory results from children assessed in the ED with cough and fever who had a CXR performed.**

| Laboratory Test | value | statistic | No pneumonia[1] (N = 139) | Bacterial pneumonia[1] (N = 62) | Viral pneumonia[1] (N = 46) | Total (N = 247) |
|---|---|---|---|---|---|---|
| White blood cell count (X10⁹/L) | | Median (IQR) | 9.1 (6.5, 13.5) | 15.6 (9.1, 19.6) | 10.1 (6.9, 15.1) | 10.0 (6.9, 15.6) |
| ESR (mm/hr) | | Median (IQR) | 20.0 (11.0, 30.0) | 45.0 (20.0, 78.0) | 29.0 (18.0, 39.0) | 24.0 (14.0, 40.0) |
| CRP (mg/L) | | Median (IQR) | 11.5 (4.7, 32.4) | 58.8 (24.8, 227.0) | 19.4 (7.6, 46.7) | 17.2 (7.0, 55.2) |
| PCT (µg/L) | | Median (IQR) | 0.1 (0.1, 0.3) | 1.0 (0.1, 7.8) | 0.2 (0.1, 1.0) | 0.1 (0.1, 1.0) |
| Positive NP swab | | n (%) | 102 (75.0) | 26 (44.1) | 37 (88.1) | 165 (69.6) |
| NP swab—first potential pathogen (N = 165)[2] | | | 102 | 26 | 37 | 165 |
| | Adenovirus | n (%) | 4 (3.9) | 1 (3.8) | 3 (8.1) | 8 (4.8) |
| | Bocavirus | n (%) | 13 (12.7) | 3 (11.5) | 3 (8.1) | 19 (11.5) |
| | Coronavirus | n (%) | 1 (1.0) | 0 (0.0) | 0 (0.0) | 1 (0.6) |
| | Enterovirus/Rhinovirus | n (%) | 20 (19.6) | 8 (30.8) | 2 (5.4) | 30 (18.2) |
| | Influenza | n (%) | 17 (16.7) | 2 (7.7) | 4 (10.8) | 23 (13.9) |
| | Metapneumovirus | n (%) | 7 (6.9) | 0 (0.0) | 2 (5.4) | 9 (5.5) |
| | Parainfluenza | n (%) | 9 (8.8) | 1 (3.8) | 5 (13.5) | 15 (9.1) |
| | RSV | n (%) | 29 (28.4) | 2 (7.7) | 18 (48.6) | 49 (29.7) |
| | *Chlamydophila pneumoniae* | n (%) | 1 (1.0) | 0 (0.0) | 0 (0.0) | 1 (0.6) |
| | *Mycoplasma pneumoniae* | n (%) | 1 (1.0) | 9 (34.6) | 0 (0.0) | 10 (6.1) |
| NP swab—second potential pathogen (N = 25) | | | 15 | 4 | 6 | 25 |
| | Adenovirus | n (%) | 3 (20.0) | 0 (0.0) | 0 (0.0) | 3 (12.0) |
| | Bocavirus | n (%) | 0 (0.0) | 2 (50.0) | 0 (0.0) | 2 (8.0) |
| | Coronavirus | n (%) | 0 (0.0) | 1 (25.0) | 0 (0.0) | 1 (4.0) |
| | Enterovirus/Rhinovirus | n (%) | 9 (60.0) | 0 (0.0) | 3 (50.0) | 12 (48.0) |
| | Metapneumovirus | n (%) | 0 (0.0) | 0 (0.0) | 1 (16.7) | 1 (4.0) |
| | Parainfluenza | n (%) | 1 (6.7) | 0 (0.0) | 0 (0.0) | 1 (4.0) |
| | RSV | n (%) | 2 (13.3) | 0 (0.0) | 2 (33.3) | 4 (16.0) |
| | *Mycoplasma pneumoniae* | n (%) | 0 (0.0) | 1 (25.0) | 0 (0.0) | 1 (4.0) |
| NP swab—third potential pathogen (N = 2) | | | 0 | 2 | 0 | 2 |
| | Enterovirus/Rhinovirus | n (%) | | 1 (50.0) | | 1 (50.0) |
| | RSV | n (%) | | 1 (50.0) | | 1 (50.0) |
| Positive blood culture[3] | | n (%) | 3/135 (2.2) | 4/62 (6.5) | 0/45 (0.0) | 7/242 (2.9) |

CRP–C-reactive protein; CXR–chest radiograph; ED–emergency department; ESR–erythrocyte sedimentation rate; IQR–inter-quartile range; NP–nasopharyngeal;

PCT–procalcitonin; RSV -respiratory syncytial virus; SD—standard deviation

[1]This diagnosis was based on the *Consensus Diagnosis* of an expert panel

[2] NP cultures were all negative for pneumococcus, *Haemophilus* species, *Staphylococcus aureus*, group A streptococcus, *Moraxella catarrhalis*, and *Bordetellae*

[3]Blood cultures were positive for *Streptococcus pneumoniae* (n = 5 (2 "Bacterial Pneumonia/ 3 "No Pneumonia"), *Staphylococcus aureus* (N = 1 ("Bacterial Pneumonia") and Gram-negative bacilli (N = 1 ("Bacterial Pneumonia"))

## Panel's clinical, ED and full data assessments compared to consensus diagnosis

The 247 cases were each reviewed by seven panelists which yielded 1729 individual decisions in each of the three data waves. With dichotomization of the panelists' diagnosis into bacterial pneumonia or not (combining no or viral pneumonia), panelists changed their diagnosis 61 of 1729 times (3.5%) between their *Immediate Clinical Assessment* and their *Full ED Assessments*, and an additional 28 of 1729 times (1.6%) between their *Full ED* and *Full Follow-up Assessments*. Consensus Diagnosis was reached with no "forcing" (i.e. all seven panelists independently made the same diagnosis) for 73 of the 247 cases (30%). The agreement between the majority of the Panelists' *Immediate Clinical Assessment* (based on history and physical exam only) and the Panel's *Consensus Diagnosis* was low (Kappa 0.30 (0.18, 0.41)). However, the majority of the Panelists' *Full ED Assessment* (adding CBC and viewing of the CXR) showed substantial agreement (Kappa 0.76 (0.68, 0.84)), as did the Panelists' *Full Follow-up Assessment* with the Panel's *Consensus Diagnosis* (Kappa .93 (0.89, 0.88)).

The panelists' degree of certainty on a scale from 0 to 100 for their: *Immediate Clinical Assessment* was median 35 (IQR 30–45); *Full ED assessment* was median 50 (IQR 43–60); *Full Follow-up Assessment* was median 62 (IQR 52–72); and *Consensus Diagnosis* was median 79 (IQR 73–84).

## ED versus panel consensus diagnosis

The diagnosis of typical bacterial pneumonia was considerably higher for ED physicians (126/247 (51%)) than the Panel's *Consensus Diagnosis* (44/247 (18%)) (Table 3). The agreement between a diagnosis of bacterial pneumonia by the ED physician and the Panel's *Consensus Diagnosis* was low (Kappa 0.15 (0.08, 0.21)). The Sankey Diagram (Fig 7) shows the most common specific discrepancy was an ED diagnosis of viral pneumonia when the *Consensus Diagnosis* was "not pneumonia".

## Accuracy of individual demographics, symptoms, signs, and laboratory tests

The optimal cut-off values for ESR, CRP and PCT for diagnosis of the types of pneumonia are shown in Table 4. The sensitivity of laboratory cut-offs ranged from 35% to 50% for a diagnosis of pneumonia, but increased to 50% to 64% and 66% to 87% for a diagnosis of bacterial and typical bacterial pneumonia, respectively. A WBC count > 15 X $10^9$/ was 67% sensitive and 83% specific for typical bacterial pneumonia.

The odds ratio of demographic, clinical and laboratory features for predicting the presence of pneumonia, bacterial pneumonia and typical bacterial pneumonia are shown in Tables 5–7 respectively. WBC, ESR, CRP and PCT were consistently predictive across all three categories

**Table 3. Differences in frequency and percentage for ED physician diagnoses versus panel consensus diagnosis for children assessed in the ED with cough and fever who had a CXR performed.**

| Pneumonia Classification | ED MD diagnosis n (%) | Consensus diagnosis n (%) |
|---|---|---|
| Not Pneumonia | 108 (43.7) | 139 (56.3) |
| Atypical Bacteria | 3 (1.2) | 18 (7.3) |
| Typical Bacteria | 126 (51.0) | 44 (17.8) |
| Viral pneumonia | 10 (4.0) | 46 (18.6) |

CXR–chest radiograph; ED–emergency department; MD–medical doctor

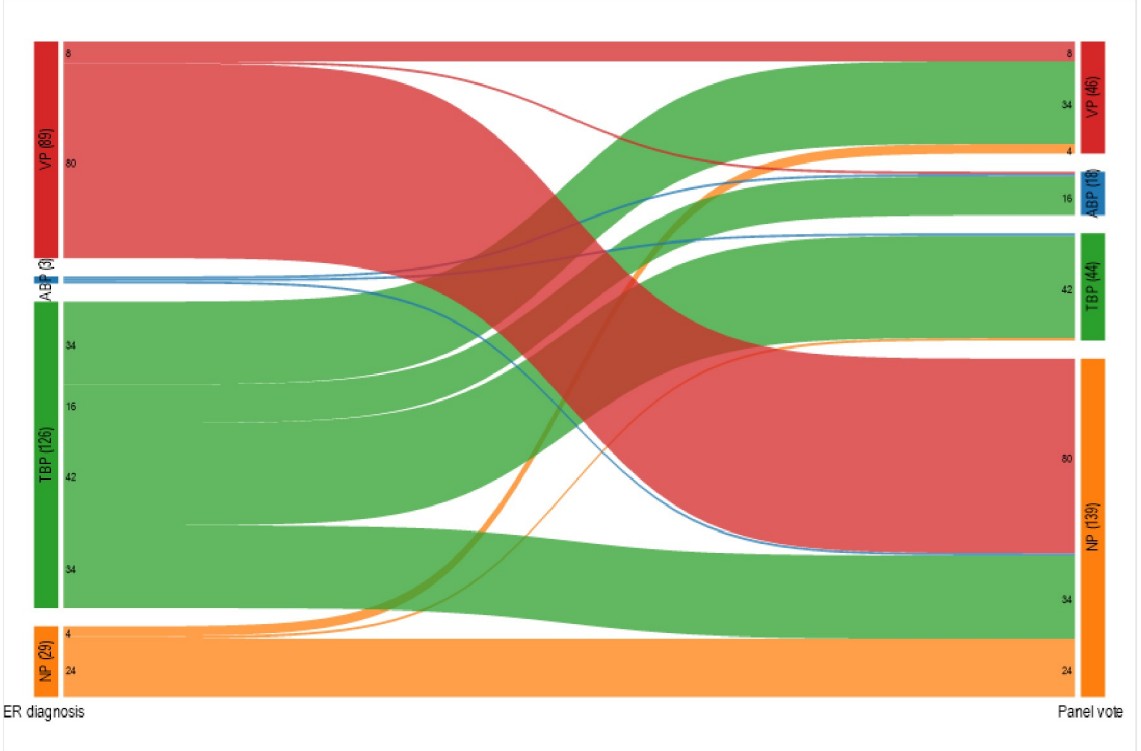

**Fig 7. Comparison of ED physician to panel consensus diagnosis for 247 children assessed in the ED with cough and fever who had a CXR performed for possible pneumonia.** NP = No Pneumonia; VP = Viral Pneumonia; TBP = Typical Bacterial Pneumonia; ABP = Atypical Bacterial Pneumonia; ER diagnosis = diagnosis by site treating ED physician; Panel vote = Expert Panel's Consensus Diagnosis.

of pneumonia with ORs for bacterial pneumonia ranging from 4 up to 11. Cough for 5 or more days was predictive for both pneumonia (Table 5) and bacterial pneumonia (Table 6). The presence of nasal flaring was predictive for pneumonia (Table 5). Age 6–16 years was predictive for bacterial pneumonia (Table 6). Absence of wheeze was a significant predictor of bacterial pneumonia (Table 6) when documented by a nurse but not when documented by a physician and, in contrast, absence of wheeze was a significant predictor for typical bacterial pneumonia when documented by a physician but not when documented by a nurse (Table 7).

## Discussion

For children seen in the ED with fever and cough for whom a diagnosis of pneumonia was suspected, a consensus panel concluded that only one quarter had bacterial pneumonia in contrast to ED physicians who diagnosed more than half with bacterial pneumonia. One potential explanation for the discrepancy would be that ED physicians most often made decisions prior to knowing microbiologic results or the radiologists' opinion of the CXR. Their main goal is not to miss a diagnosis of typical bacterial pneumonia as this could result in harm, leading them to err on the side of making this diagnosis. ED physicians achieved this goal as they diagnosed bacterial pneumonia for all but one of 62 cases that the panelists considered to be bacterial pneumonia. The most common discordance was diagnosis of viral pneumonia by the ED physician when the panel consensus diagnosis was "not pneumonia"; this would have limited consequences as antibiotics are not beneficial for either diagnosis.

**Table 4. Laboratory features that predicted pneumonia diagnosis for children assessed in the ED with cough and fever who had a CXR performed.**

| | | N | Sensitivity % (n/N) | Specificity % (n/N) | PPV % (n/N) | NPV % (n/N) |
|---|---|---|---|---|---|---|
| Pneumonia versus not pneumonia | WBC >15 vs ≤15 X10$^9$/L | 238 | 40.4(42/104) | 84.3(113/134) | 66.7(42/ 63) | 64.6(113/175) |
| | WBC<4 vs 4–15 X10$^9$/L | 175 | 3.3(4/62) | 95(108/113) | 44.4(4/9) | 65.1(108/166) |
| | WBC>15 vs 4–15 X10$^9$/L | 229 | 42.0(42/100) | 83.7(108/129) | 66.7(42/63) | 65.1(108/166) |
| | ESR≥47 vs <47 mm/hr | 221 | 35.1(33/ 94) | 92.9(118/127) | 78.6(33/42) | 65.9(118/179) |
| | CRP> = 42 vs <42 mg/L | 231 | 50.0(49/ 98) | 80.5(107/133) | 65.3(49/ 75) | 68.6(107/156) |
| | PCT> = 0.85 vs<0.85 ng/mL | 225 | 43.4(43/ 99) | 84.9(107/126) | 69.4(43/ 62) | 65.6(107/163) |
| Bacterial pneumonia versus viral pneumonia or no pneumonia | WBC >15 vs ≤15 X10$^9$/L | 238 | 50.8(31/ 61) | 81.9(145/177) | 49.2(31/ 63) | 82.9(145/175) |
| | WBC<4 vs 4–15 X10$^9$/L | 175 | 3.3(1/30) | 84.5(137/145) | 11.1(1/9) | 82.5(137/166) |
| | WBC>15 vs 4–15 X10$^9$/L | 229 | 51.7(31/ 60) | 81.1(137/169) | 49.2(31/ 63) | 82.5(137/166) |
| | ESR≥47 vs <47 mm/hr | 221 | 50.0(28/ 56) | 91.5(151/165) | 66.7(28/ 42) | 84.4(151/179) |
| | CRP> = 42 vs <42 mg/L | 231 | 64.3(36/56) | 77.7(136/175) | 48.0(36/75) | 87.2(136/156) |
| | PCT> = 0.85 vs<0.85 ng/mL | 225 | 52.6(30/57) | 81.0(136/168) | 48.4(30/62) | 83.4(136/163) |
| Typical bacterial pneumonia versus atypical pneumonia, viral pneumonia or no pneumonia | WBC >15 vs ≤15 X10$^9$/L | 238 | 67.4(29/43) | 82.6(161/195) | 46.0(29/63) | 92.0(161/175) |
| | WBC<4 vs 4–15 X10$^9$/L | 175 | 7.1(1/14) | 95.0(153/161) | 11.1(1/9) | 92.2(153/166) |
| | WBC>15 vs 4–15 X10$^9$/L | 229 | 69.0(29/ 42) | 81.8(153/187) | 46.0(29/ 63) | 92.2(153/166) |
| | ESR≥47 vs <47 mm/hr | 221 | 65.8(25/38) | 90.7(166/183) | 59.5(25/42) | 92.7(166/179) |
| | CRP > = 42 vs <42 mg/L | 231 | 86.8(33/38) | 78.2(151/193) | 44.0(33/75) | 96.8(151/156) |
| | PCT> = 0.85 vs<0.85 ng/mL | 225 | 75.0(30/ 40) | 82.7(153/185) | 48.4(30/ 62) | 93.9(153/163) |

CRP—C-reactive protein; ESR–erythrocyte sedimentation rate; NPV–negative predictive value; PCT–procalcitonin; PPV–positive predictive value

The individual clinical symptoms and examination findings that predicted bacterial pneumonia, albeit with low accuracy, were older age (6 to 16 years of age), cough for more than five days and absence of wheeze as assessed by the study nurse. However, it is of note that the majority of children with bacterial pneumonia were 5 years of age or younger. The significance of wheeze detected by the nurse versus the physician is unclear as wheeze can be intermittent. A systematic review of predictors of radiographic pneumonia in children found that presence of symptoms for more than 3 days (including cough) had a significant positive likelihood ratio, but absence of wheeze did not [5]; this review did not report on age as a predictor. Two studies which derived clinical scores for predicting bacterial or radiographic pneumonia found both age and absence of wheeze to be important predictors and included them in diagnostic algorithms [15, 16]. In contrast to two published studies and a systematic review [4, 5, 17], we found that neither the presence of fever in the ED nor history of persistent fever predicted bacterial pneumonia. It is important to note, however, that fever on admission just missed achieving statistical significance (p = 0.0524). The borderline significance of this finding may be due to our inclusion criteria requiring children to have either a history or presence of fever to be enrolled.

Laboratory tests, specifically elevated WBC, ESR, CRP and PCT, were significantly better at predicting bacterial pneumonia than were signs and symptoms, with area under ROC of between 0.60 and 0.65. A systematic review of studies which evaluated biomarkers as predictors of bacterial pneumonia found similar area under ROC for each of these tests to the current study [7]. While substantially more accurate than clinical findings, individual laboratory results were not sufficiently accurate to serve as a surrogate reference standard. Our data also show that detection of viruses does not exclude bacterial pneumonia as almost one-third of those with bacterial pneumonia had viral coinfection.

**Table 5. Demographic, clinical and laboratory features that predicted pneumonia versus no pneumonia for children assessed in the ED with cough and fever who had a CXR performed.**

| | OR with 95%CI | Area under ROC with 95% CI | Pr > Chi-Square |
|---|---|---|---|
| Male gender | 1.130(0.682–1.872) | 0.5152(0.4523–0.5780) | 0.6348 |
| Age 1–5 years (versus < 1 year or 6–17 years) | 0.797(0.474–1.342) | 0.5264(0.4654–0.5874) | 0.3936 |
| Age 6–16 years (versus <6 years) | 1.134(0.638–2.015) | 0.5120(0.4567–0.5672) | 0.6690 |
| Admission temperature ≥38˚Celsius | 1.449(0.874–2.404) | 0.5460(0.4833–0.6087) | 0.1506 |
| Fever for ≥ 5 days versus for a shorter duration | 1.152(0.656–2.024) | 0.5140(0.4577–0.5703) | 0.6230 |
| **Cough for ≥ 5 days versus for a shorter duration** | **2.009(1.204–3.352)** | **0.5861(0.5241–0.6482)** | **0.0075** |
| Oxygen saturation on room air < 93% versus higher | 0.665(0.364–1.214) | 0.5372(0.4818–0.5925) | 0.1842 |
| Crackles recorded on physical examination by RN | 1.848(1.093–3.122) | 0.5711(0.5105–0.6318) | 0.0218 |
| Wheeze recorded on physical examination by RN | 0.864(0.406–1.839) | 0.5082(0.4661–0.5502) | 0.7049 |
| Prolonged expiration recorded on physical examination by RN | 0.897(0.409–1.968) | 0.5056(0.4652–0.5460) | 0.7864 |
| Crackles recorded on physical examination by MD | 1.487(0.862–2.565) | 0.5488(0.4819–0.6158) | 0.1540 |
| Wheeze recorded on physical examination by MD | 0.608(0.288–1.284) | 0.5404(0.4811–0.5998) | 0.1921 |
| Prolonged expiration recorded on physical examination by MD | 0.986(0.476–2.041) | 0.5010(0.4508–0.5511) | 0.9694 |
| Supracostal indrawing recorded on physical examination by MD | 2.007(0.996–4.044) | 0.5529(0.4996–0.6062) | 0.0514 |
| **Nasal flaring recorded on physical examination by MD** | **3.813(1.439–10.099)** | **0.5606(0.5178–0.6035)** | **0.0071** |
| Intercostal indrawing recorded on physical examination by MD | 1.719(0.959–3.079) | 0.5583(0.4955–0.6212) | 0.0688 |
| Grunting recorded on physical examination by MD | 1.801(0.814–3.986) | 0.5346(0.4876–0.5815) | 0.1464 |
| Decreased air entry recorded on physical examination by MD | 1.513(0.878–2.607) | 0.5511(0.4842–0.6180) | 0.1359 |
| White blood cell count < 4 versus 4–15 X $10^9$/L | 1.490(0.385–5.763) | 0.5101(0.4739–0.5464) | 0.5637 |
| **White blood cell count > 15 versus 4–15 X $10^9$/L** | **3.724(2.017–6.877)** | **0.6286(0.5704–0.6868)** | **< .0001** |
| **ESR ≥ 47 mm/hr versus lower** | **7.093(3.189–15.773)** | **0.6401(0.5867–0.6935)** | **< .0001** |
| **CRP ≥ 42 mg/L versus lower** | **4.115(2.296–7.375)** | **0.6523(0.5921–0.7124)** | **< .0001** |
| **PCT ≥ 0.85 ng/mL versus lower** | **4.323(2.304–8.112)** | **0.6418(0.5835–0.7000)** | **< .0001** |

CRP—C-reactive protein; CXR–chest radiograph; ED–Emergency Department; ESR–erythrocyte sedimentation rate; MD–medical doctor; RN–Registered Nurse; PCT–procalcitonin

We used expert panel consensus as our reference standard as there is no non-invasive investigation that definitively proves the etiology of pediatric pneumonia [4]. Although it is novel to use expert panel consensus as a reference standard for pediatric bacterial pneumonia, expert panels have been used increasingly in psychiatric, cardiovascular, and other respiratory conditions [8–11]. For the diagnosis of dementia, the diagnostic accuracy of an expert panel has been shown to be superior to individual clinicians and validated as comparable to the reference standard of neuropathological based diagnosis [11]. Expert panel consensus has also been demonstrated to have high reproducibility in infectious diseases [8]. Our findings also provide face validity as panelists' assessments made after the provision of more patient data showed increasingly better agreement with the panel's Consensus Diagnosis, and panelists' diagnostic certainty increased with achievement of consensus.

A strength of our study is the novel use of an expert panel for the diagnosis of pneumonia and that the expert panel represented a heterogenous group of subspecialists each with their unique expertise relevant to the diagnosis of pneumonia in children. Notwithstanding the proven utility of expert panels, the most important limitation is that the use of our panel's Consensus Diagnosis cannot be directly validated, and it is at potential risk for incorporation bias. A second limitation is that our inclusion criteria required children to have fever and their ED physician to have ordered a chest radiograph to be enrolled. These requirements may have narrowed our cohort in ways that may have modified the predictive potential of clinical factors

**Table 6. Demographic, clinical and laboratory features that predicted bacterial pneumonia versus other types of pneumonia or "not pneumonia" for children assessed in the ED with cough and fever who had a CXR performed.**

| | OR with 95%CI | Area under ROC with 95%CI | Pr >Chi-Square |
|---|---|---|---|
| Male gender | 1.641(0.909–2.962) | 0.5605(0.4899–0.6310) | 0.1000 |
| **Age 1–5 years (versus less than 1 year or 6–17 years)** | **0.367(0.203–0.661)** | **0.6201(0.5493–0.6909)** | **0.0009** |
| **Age 6–16 years (versus <6 years)** | **3.903(2.095–7.2720** | **0.6420(0.5734–0.7106)** | **< .0001** |
| Admission temperature ≥38˚Celsius | 1.370(0.769–2.438) | 0.5391(0.4669–0.6114) | 0.2851 |
| Fever for ≥ 5 days versus for a shorter duration | 1.261(0.670–2.373) | 0.5235(0.4575–0.5894) | 0.4719 |
| **Cough for ≥ 5 days versus for a shorter duration** | **1.869(1.035–3.374)** | **0.5767(0.5061–0.6472)** | **0.0379** |
| Oxygen saturation on room air < 93% versus higher | 1.530(0.733–3.195) | 0.5362(0.4772–0.5952) | 0.2578 |
| Crackles recorded on physical examination by RN | 1.247(0.691–2.251) | 0.5259(0.4554–0.5965) | 0.4630 |
| **Wheeze recorded on physical examination by RN** | **0.274(0.080–0.932)** | **0.5542(0.5166–0.5918)** | **0.0382** |
| Prolonged expiration recorded on physical examination by RN | 0.441(0.147–1.323) | 0.5353(0.4958–0.5748) | 0.1441 |
| Crackles recorded on physical examination by MD | 1.034(0.556–1.925) | 0.5042(0.4269–0.5815) | 0.9149 |
| Wheeze recorded on physical examination by MD | 0.411(0.149–1.129) | 0.5632(0.5027–0.6237) | 0.0846 |
| Prolonged expiration recorded on physical examination by MD | 0.850(0.361–2.004) | 0.5108(0.4548–0.5669) | 0.7107 |
| Supracostal indrawing recorded on physical examination by MD | 0.977(0.442–2.161) | 0.5017(0.4412–0.5623) | 0.9549 |
| Nasal flaring recorded on physical examination by MD | 1.044(0.389–2.801) | 0.5021(0.4534–0.5508) | 0.9311 |
| Intercostal indrawing recorded on physical examination by MD | 0.791(0.400–1.564) | 0.5246(0.4541–0.5952) | 0.5011 |
| Grunting recorded on physical examination by MD | 1.143(0.474–2.755) | 0.5080(0.4534–0.5627) | 0.7662 |
| Decreased air entry recorded on physical examination by MD | 1.724(0.908–3.272) | 0.5661(0.4903–0.6419) | 0.0958 |
| White blood cell count < 4 versus 4–15 X $10^9$/L | 0.591 (0.071–4.905) | 0.5109(0.4733–0.5485) | 0.6258 |
| **White blood cell count > 15 versus 4–15 X $10^9$/L** | **4.577(2.423–8.644)** | **0.6637(0.5934–0.7340)** | **< .0001** |
| **ESR ≥ = 47 mm/hr versus lower** | **10.783(5.054–23.005)** | **0.7076(0.6381–0.7770)** | **< .0001** |
| **CRP ≥ 42 mg/L versus lower** | **6.277(3.269–12.051)** | **0.7100(0.6395–0.7805)** | **< .0001** |
| **PCT ≥ 0.85 ng/mL versus lower** | **4.722(2.473–9.019)** | **0.6679(0.5961–0.7398)** | **< .0001** |

CRP—C-reactive protein; CXR–chest radiograph; ED–Emergency Department; ESR–erythrocyte sedimentation rate; MD–medical doctor; RN–Registered Nurse: PCT–procalcitonin

such as fever. A third limitation is that our focus on pediatric emergency departments restricts the applicability of our findings to care provided in general emergency departments. A fourth limitation is that because of the relatively small number of children enrolled and reported on (n = 247) as compared with those who met study eligibility (n = 1,294) and that we do not have demographic and clinical details of those not enrolled, it is possible that our cohort of children does not accurately represent the full range of all children suspected of having pneumonia.

In conclusion, using expert panel Consensus Diagnosis as a reference standard, we determined that pediatric ED physicians significantly over diagnosed bacterial pneumonia, and that

**Table 7. Demographic, clinical and laboratory features that predicted typical bacterial pneumonia versus other types of pneumonia or "not pneumonia" for children assessed in the ED with cough and fever who had a CXR performed.**

|  | OR with 95%CI | Area under ROC with 95%CI | Pr > Chi-Square |
|---|---|---|---|
| Male gender | 1.633(0.833–3.201) | 0.5596(0.4798–0.6393) | 0.1532 |
| Age 1–5 years (versus less than 1 year or 6–17 years) | 0.811(0.417–1.579) | 0.5247(0.4442–0.6053) | 0.5377 |
| Age 6–16 years (versus < 6 years) | 1.897(0.946–3.802) | 0.6420(0.5734–0.7106) | 0.0712 |
| Admission temperature ≥38˚Celsius | 1.926(0.993–3.734) | 0.5812(0.5002–0.6622) | 0.0524 |
| Fever for ≥ 5 days versus for a shorter duration | 1.009(0.485–2.098) | 0.5009(0.4276–0.5742) | 0.9806 |
| Cough for ≥ 5 days versus for a shorter duration | 1.459(0.753–2.825) | 0.5467(0.4655–0.6278) | 0.2629 |
| Oxygen saturation on room air < 93% versus higher | 2.639(0.983–7.081) | 0.5719(0.5134–0.6304) | 0.0540 |
| Crackles recorded on physical examination by RN | 0.883(0.445–1.752) | 0.5143(0.4360–0.5925) | 0.7214 |
| Wheeze recorded on physical examination by RN | 0.275(0.063–1.195) | 0.5512(0.5116–0.5908) | 0.0850 |
| Prolonged expiration recorded on physical examination by RN | 0.498(0.144–1.726) | 0.5299(0.4858–0.5741) | 0.2716 |
| Crackles recorded on physical examination by MD | 0.737(0.367–1.477) | 0.5380(0.4504–0.6257) | 0.3894 |
| **Wheeze recorded on physical examination by MD** | **0.109(0.014–0.830)** | **0.6061(0.5577–0.6545)** | **0.0323** |
| Prolonged expiration recorded on physical examination by MD | 0.370(0.107–1.276) | 0.5535(0.5022–0.6048) | 0.1155 |
| Supracostal indrawing recorded on physical examination by MD | 0.749(0.290–1.930) | 0.5208(0.4563–0.5853) | 0.5493 |
| Nasal flaring recorded on physical examination by MD | 1.679(0.616–4.581) | 0.5281(0.4666–0.5895) | 0.3114 |
| Intercostal indrawing recorded on physical examination by MD | 0.828(0.385–1.782) | 0.5199(0.4403–0.5995) | 0.6290 |
| Grunting recorded on physical examination by MD | 1.880(0.764–4.630) | 0.5422(0.4736–0.6108) | 0.1697 |
| Decreased air entry recorded on physical examination by MD | 1.565(0.760–3.223) | 0.5544(0.4684–0.6405) | 0.2243 |
| White blood cell count < 4 versus 4–15 X $10^9$/L | 1.471(0.171–12.688) | 0.5109(0.4389–0.5829) | 0.7255 |
| **White blood cell count > 15 versus 4–15 X $10^9$/L** | **10.038(4.731–21.302)** | **0.7543(0.6783–0.8303)** | **< .0001** |
| **ESR ≥ = 47 mm/hr versus lower** | **18.778(8.142–43.307)** | **0.7825(0.7032–0.8618)** | **< .0001** |
| **CRP ≥ 42 mg/L versus lower** | **23.722(8.721–64.529)** | **0.8254(0.7636–0.8872)** | **< .0001** |
| **PCT ≥ 0.85 ng/mL versus lower** | **14.344(6.376–32.268)** | **0.7885(0.7153–0.8618)** | **< .0001** |

CRP—C-reactive protein; CXR–chest radiograph; ED–Emergency Department; ESR–erythrocyte sedimentation rate; MD–medical doctor; RN–Registered Nurse; PCT–procalcitonin

the most accurate individual patient findings for diagnosing bacterial pneumonia were greater than five years of age and cough for 5 or more days. Further we also demonstrated that standard inflammatory biomarkers were substantially more accurate than were clinical findings in the diagnosis of pediatric bacterial pneumonia.

## Supporting information

**S1 File. Alberta Children's Hospital ethics approval.**
(PDF)

**S2 File. Hospital for Sick Children's ethics approval.**
(PDF)

**S3 File. Queen's University ethics approval.**
(PDF)

**S4 File. Western University ethics approval.**
(PDF)

**S5 File. Montreal Children's Hospital ethics approval.**
(PDF)

**S6 File. Centre Hospitalier Universitaire Sainte-Justine ethics approval.**
(PDF)

**S7 File. Stollery Children's Hospital ethics approval.**
(PDF)

## Acknowledgments

This study was conducted with the assistance of Pediatric Emergency Research Canada (PERC).

## Author Contributions

**Conceptualization:** Joan L. Robinson, James D. Kellner, Martin Pusic, Martin Reed, Tim Lynch, David W. Johnson.

**Data curation:** Joan L. Robinson, Jennifer Crotts, Gabino Travassos, Guanmin Chen, Ravi Bhargava, David W. Johnson.

**Formal analysis:** Joan L. Robinson, Jennifer Crotts, Guanmin Chen.

**Funding acquisition:** James D. Kellner, Tim Lynch, David W. Johnson.

**Investigation:** Gabino Travassos, Guanmin Chen, David W. Johnson.

**Methodology:** Guanmin Chen, Tim Lynch, David W. Johnson.

**Project administration:** Jennifer Crotts, Gabino Travassos, Maala Bhatt, Kathy Boutis, Sarah Curtis, Serge Gouin, Tim Lynch, Richard van Wylick, David W. Johnson.

**Resources:** Jennifer Crotts, Gabino Travassos, Maala Bhatt, Kathy Boutis, Sarah Curtis, Serge Gouin, Tim Lynch, Richard van Wylick, David W. Johnson.

**Software:** Gabino Travassos.

**Supervision:** Tim Lynch, David W. Johnson.

**Validation:** Joan L. Robinson, James D. Kellner, Valerie G. Kirk, Martin Pusic, Martin Reed, Sarah Reid, Michael Weinstein.

**Visualization:** Joan L. Robinson, Guanmin Chen, Ravi Bhargava, David W. Johnson.

**Writing – original draft:** Joan L. Robinson, David W. Johnson.

**Writing – review & editing:** James D. Kellner, Jennifer Crotts, Gabino Travassos, Guanmin Chen, Valerie G. Kirk, Martin Pusic, Martin Reed, Sarah Reid, Michael Weinstein, Ravi Bhargava, Maala Bhatt, Kathy Boutis, Sarah Curtis, Serge Gouin, Tim Lynch, Richard van Wylick.

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
