## [Decision Letter · Decision Letter 0]

31 May 2024

PONE-D-24-14952Accuracy of the diagnosis of pneumonia in the pediatric Emergency Department: a prospective cohort studyPLOS ONE

Dear Dr. Robinson,

Thank you for submitting your manuscript to PLOS ONE and apologies for delays in finding reviewers. After careful consideration, we feel that it has merit but does not fully meet PLOS ONE’s publication criteria as it currently stands. Therefore, we invite you to submit a revised version of the manuscript that addresses the points raised during the review process.

Kind regards,

Stephen Michael Graham, FRACP, PhD

Academic Editor

PLOS ONE

“Funding for Pneumonia Prospective Cohort Study

• Canadian Institutes for Health Research Team Grant, Funded from April 2006-March 2011, Grant title: ‘Improving outcomes for ill and injured children in emergency departments’. Principal Investigator – Terry Klassen; Co-Investigators (and pneumonia project leads) – David Johnson and Tim Lynch. Can$4.8 million funded seven large multi-centre projects, one of which focused on three distinct pneumonia studies (systematic review, practice variation study and prospective cohort study)

• Alberta Children’s Hospital Foundation project grant, Funded from April 2009-April 2010, Grant title: ‘Accuracy of metabolomics for diagnosing pediatric pneumonia’. Principal Investigator – Jim Kellner; Co-Investigator – David Johnson. Can$50,000

• Alberta Lung Association project grant. Funded from April 2009-April 2010, Grant title: ‘Accuracy of metabolomics for diagnosing pediatric pneumonia’. Co-investigator. Principal Investigator – Jim Kellner; Co-Investigator – David Johnson. Can $30,000.”

3. In the online submission form you indicate that your data is not available for proprietary reasons and have provided a contact point for accessing this data. Please note that your current contact point is a co-author on this manuscript. According to our Data Policy, the contact point must not be an author on the manuscript and must be an institutional contact, ideally not an individual. Please revise your data statement to a non-author institutional point of contact, such as a data access or ethics committee, and send this to us via return email. Please also include contact information for the third party organization, and please include the full citation of where the data can be found.

4. Please upload a new copy of Figures 1B, 1C, 1D, 1E and 1F as the detail is not clear. Please follow the link for more information: https://blogs.plos.org/plos/2019/06/looking-good-tips-for-creating-your-plos-figures-graphics/" https://blogs.plos.org/plos/2019/06/looking-good-tips-for-creating-your-plos-figures-graphics/

Reviewers' comments:

Reviewer's Responses to Questions

**Comments to the Author**

1. Is the manuscript technically sound, and do the data support the conclusions?

Reviewer #1: Yes

Reviewer #2: Yes

Reviewer #3: Yes

2. Has the statistical analysis been performed appropriately and rigorously? 

Reviewer #1: Yes

Reviewer #2: Yes

Reviewer #3: Yes

3. Have the authors made all data underlying the findings in their manuscript fully available?

Reviewer #1: Yes

Reviewer #2: Yes

Reviewer #3: Yes

4. Is the manuscript presented in an intelligible fashion and written in standard English?

Reviewer #1: Yes

Reviewer #2: Yes

Reviewer #3: Yes

5. Review Comments to the Author

Reviewer #1: Thanks for asking me to review this interesting paper. I have some minor comments as below:

1. Title:

My understanding is the study focused on Paed ED in high income countries. Better to include that information in title since the approach in "low income", "middle income" is somewhat different.

2. Introduction

3. Method

I am not really clear how the Expert Consensus made decisions of "no pneumonia", "bacterial pneumonia", "atypical pneumonia", "viral pneumonia". Did they use any specific cut off for CBC, CRP, PCT, ESR? Also, what changes on CXR would suggest "atypical pneumonia", "typical pneumonia", "viral pneumonia".

A common practice in ED is children > 5 years old would be covered both "bacterial pneumonia" and "atypical bacterial pneumonia". Did you record these cases in your study?

4. Results

- Table 1: please document unit that belongs to each factor. Ex: O2 sat room air (%), HR (bpm), fever (>=38) (n)

- Table 2: what is the subgroup of "no pneumonia" + blood culture positive?

How many cases NPS positive + superimposed infection?

- Table 4:

Why to choose the cut off 4, 15 for WBC? 42 for CRP? 47 for ESR? 0.85 for PCT? Any references?

Please add a foot note for abbreviations: MD, EN

5. Discussion

My main concern is how to apply the "Expert consensus" in practice as children present to ED require an urgent approach and management. Dx of pneumonia is based on clinical findings.

Reviewer #2: Thank you for opportunity to review. Well executed project with large sample and good design. The manuscript is well written. No changes to suggest. Would recommend it is ready for publication in its current form

Reviewer #3: Line 52-54: requires a reference.

Line 117-123: Mentions the parental logs; it may be useful to include how completely these were filled. What training did the parents receive?

Methods: why not also stratify the consensus diagnosis into non pneumonia cases? It may have been useful to know how many of cases which were incorrectly diagnosed as pneumonia were bronchiolitis, asthma etc.

By limiting the consensus diagnosis certainty to >50% it removes the ability of experts who still disagreed to reflect that, or to show that they were very uncertain and it means that the certainty on the censuses diagnosis of 79 is not really comparable to the others where the experts could choose between 0 and 100. It artificially inflates this certainty.

Results:

Lines 218-222: Enrolling only 269 out of 1294 eligible patients means there is room that a lot of information was missed. Were there any trends to who which children were most likely to decline consent i.e. more unwell children, first nations children, those presenting at night etc. Could these be associated with worse clinically outcomes which limited the generally applicability of your results?

Table 1 and 2: These tables are a little bit difficult to know which statistics are the best representation of data spread for each variable. Most of these characteristics and results were given to us as both medians + IQR and means +SD. If a variable is parametric just report on Mean and SD and if it is non parametric then a median and IQR. By giving us both for every characteristic it makes it more difficult for a reader to know what information is actually the most appropriate descriptor of spread for that variable.

I also would have been very interested to see a table breaking down how the expert examples at different parts of the diagnosis process compared to the ED physicians.

Table 2: I found it interesting that 30% of the bacterial pneumonias had entero/rhino present. Is this worth commenting on. I often see junior doctors falsely reassured by the presence of entero/rhino virus on an NPA.

Table 3:

You mention that this table demonstrated discrepancies, but this table actually shows how often each group (ED or Consensus) make each diagnosis, but not how often they agree and disagree. Your Sankey diagram is what shows the actual discrepancies. For the table to do this this you would need to create a 5x5 table with one Axis listing the ED MD diagnosis and the other listing Consensus diagnosis so you can see how often they were in agreement.

Results and conclusion:

I think it would be worth a more nuanced approach to conclusions than p<0.05 = absolutely significant and p>0.05 means no significance. Clinical relevance could be discussed more. i.e. lines 323-325 you state that fever was not associated with a diagnosis. Whilst true in the most rigid sense, you actually did find a trend associating with fever and the presence of typical bacterial pneumonia, however it missed out on statistical significance by a narrow margin (p=0.0524). This would have been a reasonable part of the paper to mention that statistical significance is not the same as clinical relevance. A p of 0.0524 compared to a p of 0.0499 is completely arbitrary and fever is still likely a useful factor in discriminating typical bacterial pneumonia from other causes. The same is true of wheeze, the way you talked about it with it being significant for one type of pneumonia if heard by a nurse, but the other if heard by a physician is not very clinically relevant. For both typical and overall pneumonia, absence of wheeze from nurses and physicians either reached significance or was trending close to significance. I also wondered if it was worth grouping those two findings as just wheeze heard on examination by a either study nurse or physician, as I think both professionals findings are relevant and wheeze can be an intermittent finding.

The way the over/under 5 statement is framed feels a bit incomplete. I understand that you are saying that the pretest probability of pneumonia in an over 5 year old presenting to ED is higher, but I think it would be worth at least flagging that the majority of pneumonias were in children under 5 (and possibly higher for typical pneumonia, but we were not given that information). The way this was written feels like a junior doctor reading it could think that pneumonia is less common in 5 year old, rather than what this actually says which is that 5 year olds present much more often with these symptoms in general, but a similar number of them overall (32 v 30 in your findings) will have have pneumonia.

Lastly, I think you could give ED physicians a little bit more credit. I think it is probably worth giving them some credit for the fact that of the in indiscrepancies there were almost no missed cases of TBP. Looking at the Sankey I think maybe one misdiagnosed as NP and one misdiagnosed as ABP. This means there were almost no children placed at significant risk of adverse outcome due to undertreated pneumonia, but we could be better with our antimicrobial stewardship. I actually read that as the ED physicians doing pretty good job. I am not worried about the difference between viral pneumonia and no pneumonia, as the treatment is so similar for the most part.

6. PLOS authors have the option to publish the peer review history of their article (what does this mean?). If published, this will include your full peer review and any attached files.

Reviewer #1: **Yes: **Dr Thi Kim Phuong Nguyen

Reviewer #2: No

Reviewer #3: No

---

## [Author Response · Author response to Decision Letter 0]

16 Jul 2024

A detailed Response to Reviewers and Revised Manuscript with Tracked Changes have been submitted. We believe these will address all issues identified by the reviewers and the editor. If not, we are fully open to further revisions.

---

## [Decision Letter · Decision Letter 1]

27 Aug 2024

PONE-D-24-14952R1Accuracy of the diagnosis of pneumonia in Canadian pediatric emergency departments: a prospective cohort studyPLOS ONE

Dear Dr. Robinson,

Thank you for submitting your manuscript to PLOS ONE. After careful consideration, we feel that it has merit but does not fully meet PLOS ONE’s publication criteria as it currently stands. Therefore, we invite you to submit a revised version of the manuscript that addresses the points raised during the review process.

We look forward to receiving your revised manuscript.

Kind regards,

Maurizio Balbi

Academic Editor

PLOS ONE

Journal Requirements:

Additional Editor Comments:

**
Please address the minor issues raised by Reviewers #4 and #5. 
**

Reviewers' comments:

Reviewer's Responses to Questions

**Comments to the Author**

1. If the authors have adequately addressed your comments raised in a previous round of review and you feel that this manuscript is now acceptable for publication, you may indicate that here to bypass the “Comments to the Author” section, enter your conflict of interest statement in the “Confidential to Editor” section, and submit your "Accept" recommendation.

Reviewer #3: All comments have been addressed

Reviewer #4: All comments have been addressed

Reviewer #5: All comments have been addressed

2. Is the manuscript technically sound, and do the data support the conclusions?

Reviewer #3: Yes

Reviewer #4: Yes

Reviewer #5: Yes

3. Has the statistical analysis been performed appropriately and rigorously? 

Reviewer #3: Yes

Reviewer #4: Yes

Reviewer #5: Yes

4. Have the authors made all data underlying the findings in their manuscript fully available?

Reviewer #3: Yes

Reviewer #4: Yes

Reviewer #5: Yes

5. Is the manuscript presented in an intelligible fashion and written in standard English?

Reviewer #3: Yes

Reviewer #4: Yes

Reviewer #5: Yes

6. Review Comments to the Author

Reviewer #3: Thank you for addressing my concerns regarding the clarity of statistics and my concerns regarding the interpretation of how the ED clinicians were diagnosing. In particular i found the stats easier to interpret with less extraneous information.

Reviewer #4: Thanks to the Editorial Staff for the opportunity to review this revised article. It is an incredible piece of work. The scale of which was huge. I have read the article and I agree with the discussion. I see on Page 4 ( introduction) line 2 has an error in ' per 100,00 ') . Two words are underlined on page 22 ( results). As a PEM physician I found the article relatively easy to comprehend. This is testament to the adjustments made in lieu of the previous reviewers and the balanced discussion.

Reviewer #5: The authors have adequately addressed the reviewers' comments and queries.

The only additional comment I have is that I wonder about incorporation bias when looking at the values that diagnose pyogenic bacterial CAP, especially with regards to the inflammatory markers. If the members of the expert panel (i.e., the gold standard) believe that high inflammatory markers strongly point to a diagnosis of pyogenic bacterial CAP, and incorporate that into their evaluation, it would not be surprising that these markers predict what the reference standard considers pyogenic bacterial CAP. Perhaps a comment to that effect might be warranted in the limitations?

7. PLOS authors have the option to publish the peer review history of their article (what does this mean?). If published, this will include your full peer review and any attached files.

Reviewer #3: **Yes: **Elliot Lyon

Reviewer #4: **Yes: **Michael Barrett

Reviewer #5: No

---

## [Author Response · Author response to Decision Letter 1]

3 Sep 2024

We have made the minor changes requested by the Editor and reviewers 4 and 5.

---

## [Editor Report · Decision Letter 2]

8 Sep 2024

Accuracy of the diagnosis of pneumonia in Canadian pediatric emergency departments: a prospective cohort study

PONE-D-24-14952R2

Dear Dr. Robinson,

We’re pleased to inform you that your manuscript has been judged scientifically suitable for publication and will be formally accepted for publication once it meets all outstanding technical requirements.

Kind regards,

Maurizio Balbi

Academic Editor

PLOS ONE
---

## [Editor Report · Acceptance letter]

2 Oct 2024

PONE-D-24-14952R2 

PLOS ONE

Dear Dr. Robinson, 

I'm pleased to inform you that your manuscript has been deemed suitable for publication in PLOS ONE. Congratulations! Your manuscript is now being handed over to our production team.

Kind regards, 

on behalf of

Dr. Maurizio Balbi 

Academic Editor

PLOS ONE